# Autocatalytic degradation of the extremely potent greenhouse gas SF$_6$ in basic alcoholic solution

A. Sietmann [1], P. Heinzel[2], J. Gamper[1], D. Leitner [1], L. C. Pasqualini [1], F. R. S. Purtscher[1], H. Kopacka [1], T. S. Hofer [1], A. Zemann[2] & F. Dielmann [1] ✉

Sulfur hexafluoride (SF$_6$) is a persistent, highly potent greenhouse gas that is accumulating in the atmosphere at an increasing rate, exacerbating climate change. Whereas current degradation methods rely on forcing conditions to dissociate the inert molecule and additionally require post-treatment with basic solution to obtain harmless products, we show that SF$_6$ is mineralized in a single step at ambient temperature using potassium hydroxide in water/isopropyl alcohol mixtures under UV irradiation. The reaction proceeds via an autocatalytic mechanism, where in situ-generated acetone acts as a photosensitizer, leading to the selective formation of potassium sulfite and fluoride salts. Compared to existing methods, this approach eliminates the need for extreme temperatures, hazardous catalysts, or post-treatment, providing a scalable and easy-to-apply strategy for SF$_6$ degradation.

Reducing greenhouse gas emissions is considered crucial to mitigating anthropogenic climate change and its impact on human health and our environment[1]. Sulfur hexafluoride (SF$_6$) is recognized as the most potent greenhouse gas among the industrial gases with a GWP$_{100}$ of 26,700 (relative to CO$_2$) as a result of its high radiative efficiencies and very long atmospheric lifetime[2]. Due to its unique properties, the gas is used in various industries, mainly in the energy sector in electrical distribution systems, but also in other areas such as semiconductor manufacturing, metal casting and medical applications[3]. The large-scale production and usage of SF$_6$ has led to increasing accumulation rates of SF$_6$ in the atmosphere, with emissions reaching more than 9000 tons in 2018[4,5]. Since the end of the last century, various measures have been taken to reduce SF$_6$ emissions, including the implementation of regulations, management of SF$_6$ in closed cycles with recovery and purification methods, and the development of replacement gases[6]. In conjunction with these efforts, economically viable and environmentally friendly solutions for the disposal of SF$_6$ are critical to achieving the goal of net zero emissions[7]. However, the end-of-life degradation of SF$_6$ into harmless substances is a daunting task, as the high chemical and physical inertness of the molecule, which is desirable for applications, makes it resistant to decomposition pathways.

Current industrially applied SF$_6$ degradation processes therefore operate under harsh conditions, including combustion at 1100 °C, thermal catalytic processes and non-thermal plasma methods[8]. Apart from the high energy consumption and low selectivity of these processes, the fact that the SF$_6$ decomposition products are toxic and highly corrosive makes sophisticated operational procedures necessary and poses a challenge for operational safety[9]. Neutralization of the acidic tail gases with alkaline solution is therefore an inevitable post-treatment in order to obtain harmless final products. Given this necessity, the direct conversion of SF$_6$ in a basic solution into non-hazardous products in a single step remains a sought-after yet unresolved challenge. While solution-based reactions with SF$_6$ have been intensively investigated recently[10–13], particularly with regard to the valorization of SF$_6$[14,15], low degradation rates and the need for expensive catalysts, reagents and hazardous solvents render these strategies impractical for SF$_6$ disposal (Supplementary Information (SI), Table S26). We herein disclose a fast and easy-to-apply photocatalytic process for the mineralization of SF$_6$ in the presence of KOH in mixtures of water and isopropyl alcohol at ambient temperature. Under these conditions, SF$_6$ is completely degraded and selectively converted into sulfite and fluoride salts.

[1]Institute of General, Inorganic and Theoretical Chemistry, Universität Innsbruck, Innsbruck, Austria. [2]Institute of Analytical Chemistry and Radiochemistry, Universität Innsbruck, Innsbruck, Austria. ✉e-mail: fabian.dielmann@uibk.ac.at

## Results and discussion

### $SF_6$ degradation efficiency in basic alcoholic solution: Identification of two operational methods

Spontaneous reactions between strong organic bases and $SF_6$ were recently reported[16–19], including the opportunity to facilitate such reactions by light-induced intermolecular electron transfer[20–22]. We therefore envisioned activation of the inert $SF_6$ molecule by electron photodetachment from $OH^-$ as strategy to induce $SF_6$ degradation in hydroxide solution at ambient temperature. A solubility screening indicated, that the non-polar gas is barely soluble in water, but shows good solubility in aliphatic solvents (Table 1). Short-chain aliphatic alcohols were therefore identified as promising solvents, as they can also dissolve hydroxide salts. Guided by the particularly good solubility of both $SF_6$ and KOH in isopropyl alcohol, we irradiated a solution of KOH in isopropyl alcohol pressurized with 3 bar $SF_6$ with light at different wavelength. The reaction was carried out in a sealed NMR tube using irradiation setup 1 (SI, Chapter 4.1). When the solution was exposed to light at 280 nm (220 mW), a mixture of KF and $K_2SO_3$ precipitated within minutes, whereas no reaction was observed with light at longer wavelengths (SI, Chapter 5.3). Encouraged by this result, the same setup was used for a solvent screening, which indicated complete consumption of KOH with secondary alcohols and sluggish reactions in the case of water, primary and tertiary alcohols (SI, Chapter 5.4). To better understand the role of the alcohol in the $SF_6$ degradation process, irradiation setup 2 was designed (Fig. 1b), which includes a pressure control unit (PCU) that allows for monitoring of the $SF_6$ uptake by the reaction mixture (SI, Chapter 3.9). Consistent with the preliminary investigations on the NMR tube scale, hardly any consumption of $SF_6$ was detected by the PCU when using water, primary and tertiary alcohols as solvent. By contrast, secondary alcohols showed significantly higher $SF_6$ conversion rates (Fig. 1e). The close examination of the $SF_6$ uptake time profile of the KOH/iPrOH system discloses three distinct phases. Initially, the reaction proceeds at a very slow rate, with the PCU's first valve cycle maintaining constant $SF_6$ pressure in the reaction vessel occurring 12 minutes after the start of the irradiation. At this point, a white solid begins to precipitate from the reaction mixture (Supplementary Movie 1). The $SF_6$ consumption then rapidly accelerates to reach a constant degradation rate of 0.7 g/h during Phase II. Monitoring of the $SF_6$ uptake when terminating of the irradiation after 20 minutes and after 30 minutes reveals that the reaction is complete after Phase II (SI, Chapter 6.2.5), and the subsequent decline in $SF_6$ uptake rate during Phase III is due to the saturation of the solution with $SF_6$ (SI, Chapter 5.1.2). Notably, the KOH/*sec*-BuOH system shows a similar reaction profile, but reaches a degradation rate of only 0.3 g/h (Fig. 1e). The precipitation of solids during the reaction can be prevented by adding water to the KOH/iPrOH system resulting in a biphasic system in which the inorganic salts remain dissolved in the aqueous phase (Fig. 1c, d). Compared to the KOH/iPrOH system, slightly higher $SF_6$ degradation rates of 1.1 g/h are achieved with the biphasic system (Fig. 1e). Thus, two operational methods with similar $SF_6$ decomposition rates were identified, each of which may offer distinct advantages in terms of product separation or reaction scale-up.

### Analysis of the reaction products

The reaction mixtures of the KOH/iPrOH and the $KOH/H_2O/iPrOH$ system, obtained after 2 hours of irradiation, were separated into their volatile and solid components by distillation and subjected to comprehensive analysis by several methods (SI, Chapter 6.1 and 7.1). A pH of 8.0 was measured for the aqueous solutions of the solid residues, indicating that potassium hydroxide was fully consumed and neutralized by $SF_6$. $^{19}F$ NMR spectroscopy of the volatile and the solid products confirms the presence of fluoride ions in the solid as the only $F^-$ containing species. Additionally, X-ray powder diffraction (XRD), infrared (IR) spectroscopy and capillary electrophoresis (CE) analyses identified KF and $K_2SO_3$ as the major components of the solid products. Trace amounts of $K_2S_2O_3$ were detected in the solid products of the KOH/iPrOH system by CE, whereas no $K_2S_2O_3$ was detected for the biphasic system. $^1H$ and $^{13}C$ NMR spectroscopy and gas chromatography coupled to mass spectrometry (GC-MS) analysis of the volatile products confirm the formation of diisopropyl sulfite, acetone and water. Diisopropyl sulfite is an intermediate in the mineralization process, as it gradually reacts with KOH to give $K_2SO_3$ and iPrOH, a process verified through a separate experiment (SI, Chapter 8.5). Consequently, diisopropyl sulfite accumulates in the reaction mixture as KOH is progressively consumed. Collectively, the product analysis reveals the complete consumption of KOH during $SF_6$ degradation, resulting in the quantitative formation of KF and sulfites. In line with the reduction of $S^{VI}F_6$ to sulfites $R_2S^{IV}O_3$ (R = K, iPr), one equivalent of acetone is produced via the oxidation of isopropyl alcohol, as depicted in the overall reaction equation shown in Fig. 1a.

### Computational and experimental studies reveal an autocatalytic mechanism

We propose a reaction mechanism involving two distinct photochemical processes and a light-independent redoxcatalysis, all leading to the reductive activation of $SF_6$ (Fig. 2a).

The first process entails a light-induced electron transfer from solvated $OH^-$ to $SF_6$, which is expected to occur irrespective of the type of alcohol present, albeit with a notably low quantum efficiency. Previous studies suggest that the photodetachment of an electron from aqueous hydroxide requires approximately 6.6 eV of energy[23], but solvation effects significantly lower the vertical detachment energies (VDEs)[24]. For $(H_2O)_nHO^-$ clusters, VDEs range from 4.18 eV (n = 2) to 5.72 eV (n = 5), corresponding to light wavelengths between 297 nm and 217 nm[25]. In water, $OH^-$ is typically solvated by three to four water molecules[26], while alcohols provide less efficient solvation due to their single hydrogen bond donor site. To account for these differences, density functional theory (DFT) calculations were performed to estimate the charge-transfer-to-solvent energies of the $OH^-$ and $iPrO^-$ anions when solvated by isopropyl alcohol (SI, Chapter 12). The calculated VDE for $OH^-$ (279 nm) align with the emission wavelength of the irradiation source, while the VDE for $iPrO^-$ is slightly red-shifted (289 nm). Given the significantly lower concentration of $iPrO^-$ in the reaction mixture ($pK_a(iPrOH) = 16.5$, $pK_a(H_2O) = 14$)[27,28], the light-induced electron transfer from $OH^-$ is expected to be dominant. TDDFT calculations on $SF_6\cdots OH^-$ complexes revealed electronic excitations with significant intensity at 258.6 nm (H-bonded) and 268.0 nm (O-bonded) (see SI, Fig. S68). Orbital analysis indicates that these transitions correspond to an $n(OH^-) \rightarrow \sigma^*(SF_6)$ electron transfer, supporting the feasibility of $SF_6^{\cdot-}$ radical anion formation under highly basic conditions via a photo-induced intermolecular charge transfer process.

The second photochemical process, specific to secondary alcohols, involves the ketyl radical anion acting as long-lived reductant for $SF_6$ in solution. The radiation-induced generation of the ketyl radical

**Table 1 | Solubility of $SF_6$ in different solvents at 2 bar $SF_6$ pressure and 25 °C**

| Solvent | c($SF_6$) (mmol/L) |
|---|---|
| Water | 1 |
| Toluene | 62 |
| Ethanol | 68 |
| Tetrahydrofuran | 82 |
| Isopropyl alcohol | 103 |
| *n*-Hexane | 111 |
| Isopropyl alcohol / KOH (30 g/L) | 82 |

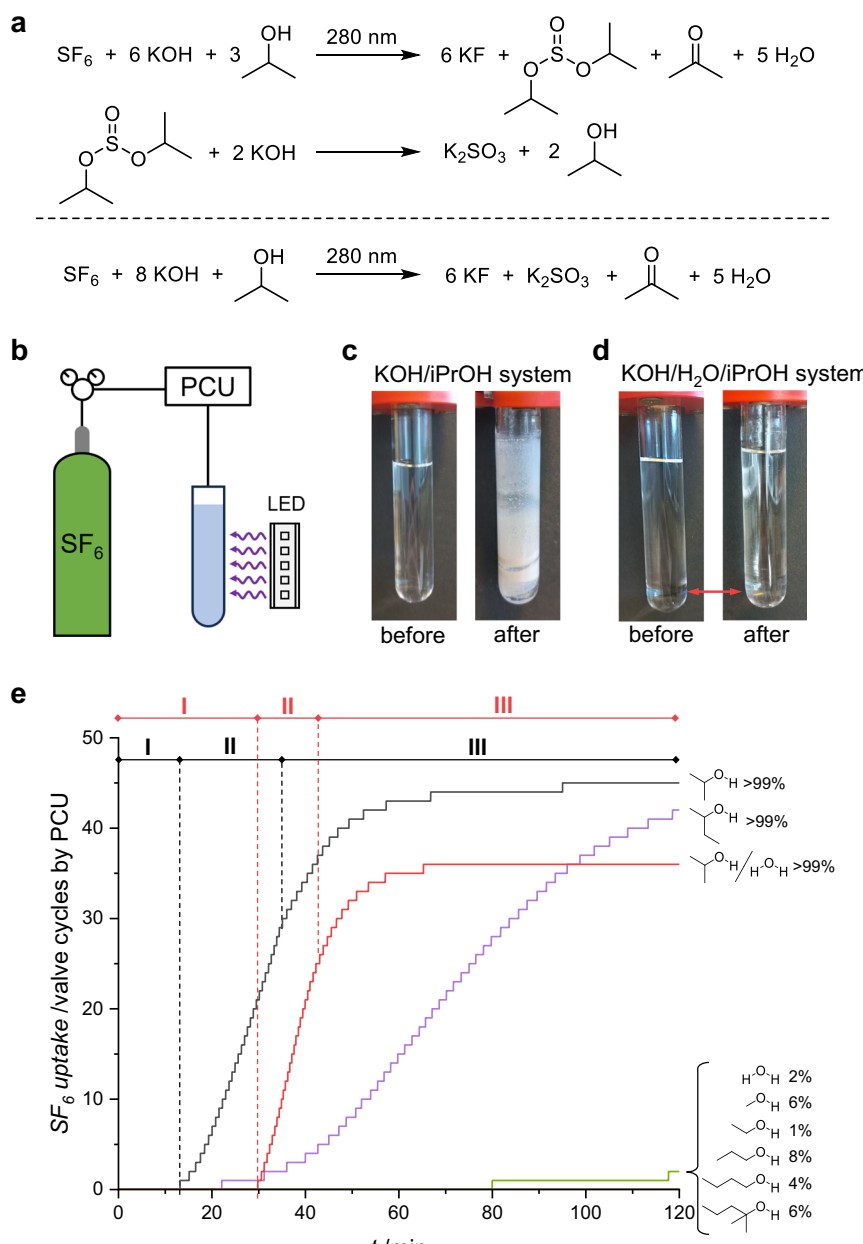

**Fig. 1 | Degradation of SF$_6$ in basic alcoholic solution. a** Overall reaction equation for the photocatalytic SF$_6$ degradation with KOH/iPrOH, **b** schematic reaction setup, **c** reaction vessel containing the KOH/iPrOH system before (left) and after (right) irradiation with light at 280 nm, **d** reaction vessel containing the biphasic KOH/H$_2$O/iPrOH system before (left) and after (right) irradiation with light at 280 nm (red arrow marks phase boundary), **e** SF$_6$ uptake time plots obtained by the PCU of different KOH/solvent systems including the KOH/iPrOH system (black line) and the biphasic KOH/H$_2$O/iPrOH system (red line) (start of irradiation at t = 0 min with initial SF$_6$ atmosphere).

anion in alkaline aqueous isopropanol solution, as part of a chain reaction, was first proposed by Scholes, Simic, and Weiss[29]. Later, Sherman postulated a radical chain mechanism for the photoreduction of nitrous oxide in alkaline isopropanol[30], establishing a precedent for the development of efficient dechlorination protocols for organochlorine compounds, including CCl$_4$[31], DDT[32] and PCBs[33]. In contrast to the organochlorine compounds, the generation of radicals by homolytic S–F photochemical bond fission is not feasible, as it requires light below 160 nm[34]. We therefore propose that the hydroxyl radicals, or O$^-$ radicals generated by deprotonation (p$K_a$(OH) = 11.9)[35], abstract a hydrogen atom from isopropanol, forming the ketyl radical[36]. Rapid deprotonation gives the ketyl radical anion (p$K_a$(Me$_2$COH) = 12.2)[37], which we detected via EPR spectroscopy due to its slow decay in solution (SI, Chapter 8.4). This anion has a more negative redox

potential (E$_{1/2}$(SCE) = −2.10 V) compared to the neutral ketyl radical (E$_{1/2}$(SCE) = −1.39 V)[38], allowing it to reduce SF$_6$ (E$_{1/2}$(SCE) = −1.90[13] or −1.80 V[39]) via electron transfer, concomitant with the formation of acetone. Note that an independent experiment confirmed that the in-situ generated dimethyl ketyl radical anion acts as the reductant toward SF$_6$ (SI, Chapter 8.8). The UV-vis spectrum of the reaction mixture post-irradiation shows the characteristic absorption band of acetone at λ$_{max}$ = 280 nm, corresponding to the n→π* transition[40]. Following rapid inter system crossing transition, which occurs with near unit efficiency[41], the excited triplet state of acetone is quenched by isopropyl alcohol via hydrogen abstraction from the α-carbon, generating two new ketyl radicals perpetuating the autocatalytic cycle.

Further experimental evidence for the autocatalytic cycle was obtained from a deuteration study. NMR analysis of the reaction

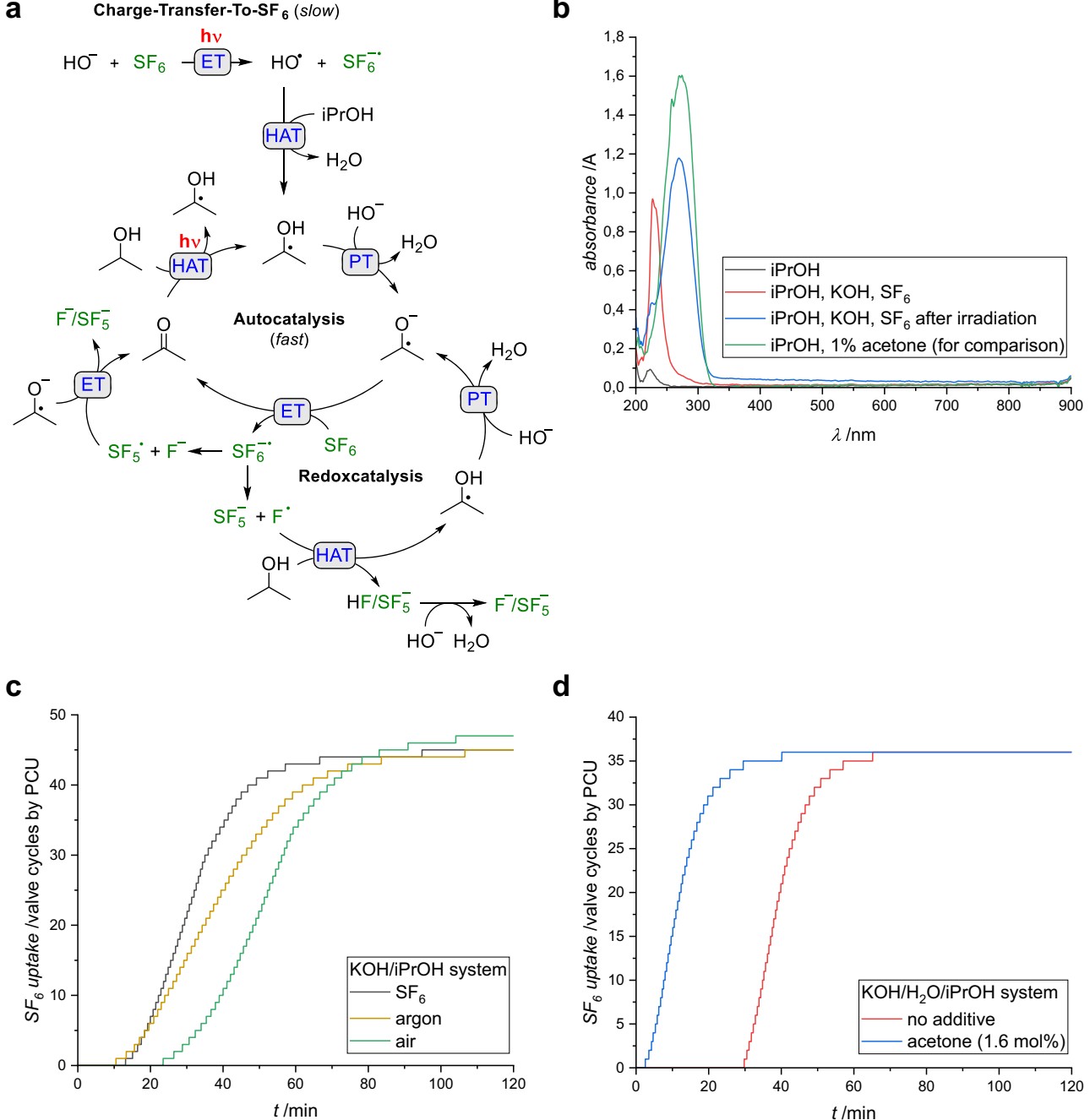

**Fig. 2 | Mechanistic studies of the SF$_6$ degradation reaction. a** Proposed mechanism for the SF$_6$ degradation using KOH in isopropyl alcohol featuring two photochemical processes and one redox cycle, **b** UV-vis absorption spectra of different solutions confirming the generation of acetone ($\lambda_{max}$ = 280 nm) during the photocatalysis, **c** SF$_6$ uptake time plots obtained by the PCU for the KOH/iPrOH system with different initial gas atmospheres (start of irradiation at $t$ = 0 min), **d** SF$_6$ uptake time plots obtained by the PCU for the KOH/H$_2$O/iPrOH system showing the influence of added acetone (start of irradiation at $t$ = 0 min).

mixture irradiated in D$_2$O instead of H$_2$O revealed the formation of HDO, isopropanol-d$_1$ (with deuterium in the hydroxy group, C$_\alpha$-position, or C$_\beta$-position), and acetone-d$_1$. The deuteration of the hydroxy group and the formation of acetone-d$_1$ are expected under the basic conditions[42]. The generation of isopropanol-d$_1$ (C$_\beta$-position) supports the proposed autocatalytic cycle, where photoexcitation of acetone-d$_1$ produces a deuterated ketyl radical. This radical undergoes hydrogen atom transfer with a second isopropanol molecule, forming isopropanol-d$_1$. Deuterium scrambling into the C$_\alpha$-position of isopropanol further proves the presence of ketyl radical intermediates and indicates that radical formation is reversible.

The light-independent redoxcatalysis is initiated by the generated SF$_6$·⁻ radical anion, which is unstable and can dissociate through two pathways: either forming a fluorine radical and an SF$_5$⁻ anion, or a fluoride anion and an SF$_5$· radical[43]. DFT calculations indicate that both pathways occurs on a comparable energetic scale, with ΔG values of 103.1 and 118.2 kJ mol⁻¹ for the dissociation into SF$_5$⁻ + F· and SF$_5$· + F⁻, respectively (SI, Chapter 12.2.2). However, hydrogen atom abstraction by F· from isopropanol is highly exergonic (ΔG = −189.1 kJ mol⁻¹), while the corresponding process for SF$_5$· is energetically unfavorable (ΔG = +50.0 kJ mol⁻¹). This suggests that the radical chain reaction is sustained only when F· is

produced, whereas the formation of $SF_5^{\cdot}$ terminates the dark reaction.

Insights into the contribution of the light-independent dark reaction to the $SF_6$ degradation process were obtained from the calculated quantum efficiency, by an intermitted irradiation experiment and by using $t$BuOO$t$Bu as a radical starter. The minimum quantum efficiency of the $SF_6$ degradation reaction was estimated to be 2.7 and 4.2 for the KOH/iPrOH and the KOH/$H_2O$/iPrOH system, respectively. These values indicate that the reaction is not solely driven by the autocatalytic cycle but also involves a redox catalysis cycle. Furthermore, significantly higher $SF_6$ degradation yields were achieved when the same total light exposure time was applied intermittently rather than continuously (94% vs. 85% yield). This confirms that the redox catalysis contributes during the dark periods, although its long-term stability is limited. Additional insight was gained by monitoring $SF_6$ degradation in the dark using tBuO$^{\cdot}$ radicals generated via thermal dissociation of $t$BuOO$t$Bu at 100 °C. This experiment revealed a turn-over number of 2.4 for the redox-catalytic $SF_6$ degradation (dark reaction) at elevated temperatures (SI, Chapter 8.7).

During Phase I, the reaction rate thus depends on the concentration of acetone, which forms gradually through the two photochemical processes. Note that a control experiment confirmed that diisopropyl sulfite does not accelerate the reaction but instead readily hydrolyses to give $K_2SO_3$ (SI, Chapter 8.5). Consistently, this induction period shortens with increased irradiation power (SI, Chapter 6.2.4) and lengthens by the presence of molecular oxygen, which inhibits the reaction by efficiently quenching the excited triplet state of acetone (Fig. 2c)[44]. Notably, introducing 1.6 mol% acetone into the reaction vessel led to the immediate uptake of $SF_6$ upon exposure to 280 nm light (Fig. 2d). Note that the temperature increase in the reaction vessel upon irradiation causes a corresponding rise in $SF_6$ pressure, which in turn causes an initial delay in the correlation between $SF_6$ uptake and valve cycles. During Phase II, $SF_6$ is consumed at a constant rate, independent of the acetone concentration in solution. This suggests that the reaction becomes either diffusion-controlled or photon-limited. A significant decrease in the reaction rate when stirring was stopped confirms that mass transport plays a role (SI, Chapter 6.2.6). During this phase, the rate of $SF_6$ consumption depends on both $SF_6$ pressure and irradiation power. However, it follows pseudo-zero-order kinetics with respect to KOH, given the large excess of KOH in solution (SI, Chapter 6.2.3). As dictated by Henry's law, the concentration of dissolved $SF_6$ is directly proportional to its partial pressure. Reducing the $SF_6$ pressure from 2 bar to 1 bar thus leads to a 50% decrease in the reaction rate (SI, Chapter 6.2.2). Similarly, if the reaction vessel initially contains an argon instead of an $SF_6$ atmosphere, the reaction proceeds more slowly due to the lower partial pressure of $SF_6$, confirming that $SF_6$ diffusion can become rate limiting (Fig. 2c). Variation of the irradiation power reveals that, for irradiation setup 2, the reaction rate remains photon-limited when fewer than 10 LEDs are used but becomes mass transfer-limited at higher irradiation powers (SI, Chapter 6.2.4). These observations confirm that both mass transport and photon flux can become rate-limiting during Phase II. This conclusion is further supported by the absence of a kinetic isotope effect when using $D_2O$ instead of $H_2O$ (SI, Chapter 8.11).

The effectiveness of secondary alcohols in the photochemical degradation of $SF_6$ is explained by their role in the autocatalytic cycle. Tertiary alcohols do not participate because they lack hydrogen atoms at the α-carbon that would allow radical anion formation. While primary alcohols could, in principle, generate such reductant, the resulting aldehyde is prone to undergoing aldol condensations or Cannizzaro-type disproportionation under the basic conditions. This is supported by the observation that irradiation of KOH solutions in primary alcohols in the presence of $SF_6$ produces yellow-to-orange solutions, with UV-vis spectra showing absorptions below 400 nm (SI, Chapter 8.3). Note that attempts to detect the organic compounds formed during this process using GC-MS or NMR spectroscopy were unsuccessful, likely due to their very low concentrations. By contrast, the KOH/iPrOH system remains colorless throughout the photochemical $SF_6$ degradation. However, the detection of traces of hexylene glycol by GC-MS analysis in the liquid phase indicates that aldol condensations do occur to some extent, though at a much slower rate than with aldehydes. Remarkably, in the biphasic KOH/$H_2O$/iPrOH system, the condensation reaction is completely suppressed due to the lower basicity of the medium. This was further confirmed by a control experiment involving prolonged stirring of acetone in both basic reaction solutions (SI, Chapter 8.6). The absence of side reactions in the biphasic system presents a significant advantage for scaling up the reaction and facilitating the reuse of the organic phase.

## Upscaling of the $SF_6$ degradation methods

To demonstrate the scalability of the $SF_6$ mineralization reaction, two additional irradiation setups (Setup 3 and 4) were designed with liquid volumes of 80 ml and 850 mL, respectively (SI, Chapter 4). In Setup 3, the reaction solution was irradiated from the top through a quartz glass window in order to prevent deposition of the inorganic salts at the light entry point during the reaction. Indeed, when using the KOH/iPrOH system, inorganic salts accumulated at the bottom of reaction vessel. However, the reaction proceeded very slowly, likely due to the limited light intensity (SI, Chapter 6.3.1). To address this limitation, Setup 4 was developed, where the reaction mixture was irradiated through a quartz tube inside a 1 L round-bottom flask, ensuring optimal irradiation efficiency (Fig. 3). Under these conditions, using the KOH/iPrOH system, 19.0 g of $SF_6$ were mineralized concomitant with complete consumption of KOH within 24 hours (Supplementary Movie 2). After the induction period, the reaction reaches a maximum decomposition rate of 5 g/h, which gradually declined as the inorganic salts accumulated on the inner quartz tube, reducing the light penetration (SI, Chapter 6.3.2). In contrast, salt precipitation is prevented in the biphasic KOH/$H_2O$/iPrOH system, ensuring constant irradiation power throughout the reaction. Therefore, $SF_6$ decomposition reaches a constant rate of 12 g/h in the diffusion-controlled Phase II regime using Setup 4. Consistent with the abovementioned results from the 15 mL-scale reaction, the $SF_6$ degradation in Phase I is delayed by the presence of oxygen but accelerated by the addition of acetone (Fig. 3).

These findings underscore the potential of the KOH/$H_2O$/iPrOH system for large-scale photochemical $SF_6$ degradation and suggest that even higher decomposition rates can be achieved with improved reactor designs that enhance mass transfer between phase interfaces. Notably, the current process already outperforms other solution-based methods by more than two orders of magnitude (see Table S26 for an overview). A key advantage of this approach is that the in-situ fragmentation and mineralization of $SF_6$ in solution prevents the formation of hazardous by-products. Instead, typical toxic and acidic impurities occurring in used $SF_6$, such as HF, $H_2S$, and $SO_2$[45], are effectively mineralized by the same reaction medium, while $N_2$ remains unreacted. This makes the process highly versatile and well-suited for treating both pure $SF_6$ gas and $SF_6$-containing gas mixtures at their end-of-life stage.

In conclusion, this study presents a fast and scalable photocatalytic method for the mineralization of the extreme greenhouse gas $SF_6$ using potassium hydroxide in isopropyl alcohol/water mixtures under ambient conditions. The reaction proceeds via an autocatalytic mechanism driven by light-induced electron transfer and radical-mediated redox processes, selectively yielding non-hazardous potassium sulfite and fluoride salts. Two operational protocols were identified, each with distinct advantages: 1) The KOH/iPrOH system facilitates straightforward product separation, as the salts precipitate from the solution. 2) The biphasic KOH/$H_2O$/iPrOH system offers enhanced efficiency and better scalability by preventing salt precipitation and maintaining consistent reaction rates. Compared to

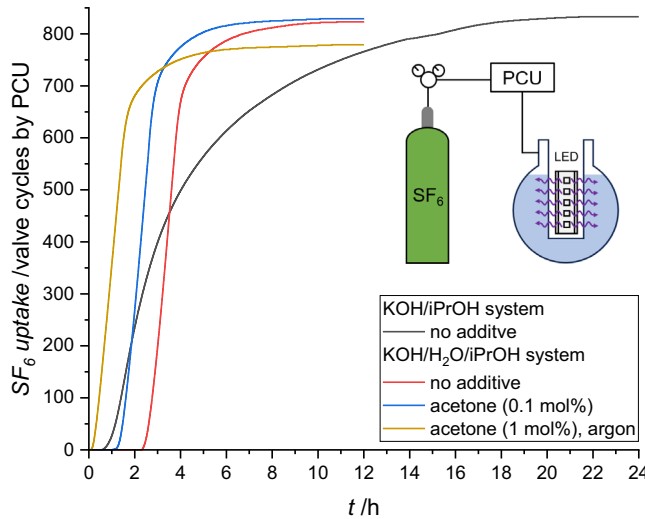

**Fig. 3 | Mineralization of 19 g of SF$_6$ in one batch.** Schematic representation of irradiation setup 4 and the SF$_6$ uptake time plots obtained by the PCU for the KOH/iPrOH system (black: initial air atm., no additive), the KOH/H$_2$O/iPrOH system (red: initial air atm., no additive; blue: initial air atm., 0.1 mol% acetone; brown: initial argon atm., 1 mol% acetone).

existing SF$_6$ degradation methods, our approach offers superior performance in terms of reaction rates, scalability, and environmental safety, eliminating the need for extreme temperatures, hazardous catalysts, or post-treatment steps. The successful upscaling of this reaction underscores its industrial potential. Future research should focus on optimizing reactor design for improved mass transfer and exploring its applicability to other persistent polyfluorinated pollutants.

## Methods

### Materials
Sulfur hexafluoride (SF$_6$) was generously donated by the company DILO GmbH. All other reagents and solvents were purchased from commercial sources and used as received if not stated otherwise. For further details, refer to the SI, Chapter 1.

### Analytical methods
Nuclear magnetic resonance (NMR) spectroscopy was performed using Bruker Avance I 400 or Bruker Avance III 400 spectrometers for $^1$H, $^{13}$C, and $^{19}$F nuclei, and a Bruker DPX 300 spectrometer for $^{33}$S NMR spectroscopy. Gas chromatography coupled to mass spectrometry was performed on a Shimadzu Nexis GC2030 gas chromatograph (GC) equipped with an autosampler, flame ionization detector, triple quadrupole mass spectrometer (MS) with an EI ion source and a Rxi-5ms crossbond 5% diphenyl/ 95% polysiloxane column (30 m length, 250 μm diameter). Infrared (IR) spectroscopy was carried out on a Bruker Alpha II FT-IR spectrometer with a Platin ATR device. X-ray powder diffractometry (XRD) was performed on a STOE Stadi P powder diffractometer. Measurements were performed in transmission geometry with Ge(111)-monochromatized MoK-L$_3$ radiation ($\lambda = 0.7093$ Å) within a range of $2\theta = 2$–$70°$, a step size of 0.015° and a Mython 1 K detector. Capillary electrophoresis (CE) was carried out on a 1600 CE-System from Agilent with a capacitively coupled contactless conductivity detector from Innovative Sensor Technologies GmbH and a capillary (80 cm length, 50 μm inner diameter) from Polymicro Technologies. pH determination was performed on an inoLab® pH 7110 meter. Ultraviolet-visible (UV-vis) spectroscopy was carried out on a PerkinElmer LAMBDA XLS+ spectrophotometer using standard quartz UV-vis cuvettes (d = 1 cm). Electron paramagnetic resonance (EPR) spectroscopy was performed on a Bruker Magnettech ESR5000

X-band spectrometer equipped with a temperature control unit and a high intensity mercury-xenon lamp (Hamamatsu 200 W Mercury Xenon 365 nm wide band L9566-06A) with a spectral distribution of 240 to 550 nm. The pressure control unit (PCU) was self-built consisting of an Arduino Nano E/A-Board with an ATmega328 microcontroller, an electrical magnetic valve and a piezo-resistive silicon pressure sensor connected via gastight tubes. For further details on the analytical methods, see the SI, Chapter 3.

### Experimental setup
Photochemical experiments were conducted in quartz vessels or glass vessels with quartz components to ensure high optical transparency. The light sources used included LEDs with wavelengths of 585 nm (VCClilte LED; Typ VAOL-SA1xAx-SA), 405 nm (EvoluChem™ LED 405PF), 365 nm (EvoluChem™ LED 365PF), 310 nm (Seoulviosys LED Typ CUD1KFMA), 280 nm (Led-Tech XBT-3535-UV LED) and a mercury-xenon lamp (MPDS-BASIC, nova®Light TXE 150). Depending on the scale of the SF$_6$ degradation reaction, the irradiation experiments were carried out using one of the following setups: Setup 1 (1 mL, NMR tube), Setup 2 (15 mL, quartz tube), Setup 3 (80 mL, 100 mL flask with a quartz looking window), or Setup 4 (850 ml, 1 L round-bottom flask with an internal quartz tube). For a detailed description of the experimental setups, refer to the SI, Chapters 2 and 4.

### Typical experimental procedures
Degradation of SF$_6$ in the KOH/iPrOH System using Irradiation Setup 2: A quartz tube was charged with a solution of potassium hydroxide (1.0 g, 16.8 mmol) in isopropanol (15 ml). The solution was degassed by four freeze-pump-thaw cycles, and the reaction tube was saturated with SF$_6$ at a pressure of 2 bar. The uptake of SF$_6$ was monitored using the pressure control unit (PCU). Once the solution was saturated with SF$_6$ gas, the reaction mixture was irradiated with light at 280 nm for 2 hours, resulting in the precipitation of inorganic salts (KF and K$_2$SO$_3$). After irradiation, the reaction mixture was transferred into a 100 mL round-bottom flask for analysis, using small amounts of water to ensure complete transfer. The volatiles were separated from the inorganic salts by evaporation under reduced pressure. The collected volatile components (11.9 g) and the remaining solid components (1.02 g) were analysed using various characterization techniques, including $^1$H, $^{19}$F, $^{13}$C{$^1$H}, and $^{33}$S NMR spectroscopy, GC-MS, IR, UV-vis, powder XRD, CE, and pH determination. For a detailed description of the product analysis, see the SI, Chapter 6.

Degradation of SF$_6$ in the KOH/H$_2$O/iPrOH System using Irradiation Setup 4: A borosilicate 1 L round-bottom flask with an internal quartz tube (diameter 34 mm) was charged with a solution of potassium hydroxide (60 g, 1.0 mol) in water (100 mL) and isopropanol (750 mL). The mixture formed a biphasic system, consisting of an upper isopropanol layer and a lower aqueous KOH solution. Vigorous stirring was applied to create an emulsion. The reaction vessel was pressurized with 2 bar SF$_6$. Once saturation with SF$_6$ was confirmed by the PCU, the emulsion was irradiated with light at 280 nm for 12 hours, with the uptake of SF$_6$ continuously monitored using the PCU. After irradiation, the two phases were separated in a separation funnel. The isopropanol phase was analysed by GC-MS and $^1$H NMR spectroscopy. An aliquot (1 mL) of the aqueous phase (134 g, 92.4 ml) was analysed by $^{19}$F quantitative NMR spectroscopy and CE. For a detailed description of the product analysis, see the SI, Chapter 7.2.

### Computational studies
To estimate the energy required to detach an electron from potential donors and to evaluate the influence of solvation on this process, the vertical detachment energies (VDE) of the donors OH$^-$, iPrO$^-$ and pure iPrOH were examined. Density functional theory (DFT) calculations were performed at four different levels of theory (B3LYP[46], ωB97XD[47], CAM-B3LYP[48], HSE06[49]) in conjunction with 6-311 + + G(d,p)[50,51] as basis

set. Energy minimizations were conducted both in vacuo and in implicit iPrOH solvent, modeled using the polarizable continuum model (PCM)[52,53]. For the estimation of VDEs, time-dependent DFT (TDDFT)[54] calculations were performed on the optimized geometries of the molecules using all four DFT methods. These TDDFT calculations were performed both in vacuo and in implicit solvent, employing two different basis sets, i.e. 6-311 + + G(d,p) and aug-cc-pVTZ[55,56]. To achieve a more accurate determination of the energy required to transfer an electron from $OH^-$ to the $SF_6$ molecule, energy minimization and subsequent TDDFT calculations were performed on a combined $SF_6 \cdots OH^-$ system. These calculations were carried out at the ωB97XD/6-311 + + G(d,p)/PCM level of theory, using tight convergence criteria for energy and gradients. Additionally, the potential dissociation pathways of $SF_6^{\bullet-}$ and subsequent hydrogen atom transfer reactions were investigated using ωB97XD/6-311 + + G(d,p)/PCM. All optimized structures were confirmed as true minima on the potential energy surface through frequency calculations, which were also used for thermochemical analysis. All calculations were performed using the Gaussian16[57] software package. For additional details on the computational studies, see the SI, Chapter 12.

## Data availability
All data are available in the main text and the Supplementary Information/Source Data. All data are available from corresponding authors upon request. Source data are provided with this paper.

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

## Acknowledgements

We thank Dr. Marius Wünsche for valuable scientific discussions. We thank Dr. Philipp Rotering for the initial conceptual design of the PCU. We thank DILO GmbH for providing a cylinder of SF$_6$ gas. This research was funded in part by the Austrian Science Fund (FWF) 10.55776/ PAT3489723 received by F.D., Prototypenförderung des Förderkreises 1669 received by F.D. and Doktoratsstipendium der Nachwuchsförder-ung der Universität Innsbruck 2021/3 Tranche received by A.S.

## Author contributions

Conceptualization of the project was by F.D. and A.S. The project was supervised and directed by F.D. Experimentation was performed by A.S. Powder diffraction studies were performed by L.C.P. EPR measurements were performed by D.L. $^{33}$S NMR measurements were performed by H.K. Capillary electrophoresis was performed by P.H. and A.Z. DFT calculations were performed by J.G., F.R.S.P., and T.H. The manuscript was written by A.S. and F.D. All authors have given approval for the final version of the manuscript.

## Competing interests

Th authors declare no competing interests.
