## [Transparent Peer Review file · Nature Communications]

Autocatalytic degradation of the extremely potent greenhouse gas SF₆ in basic alcoholic solution

Corresponding Author: Dr Fabian Dielmann

Version 0:

Reviewer comments:

Reviewer #1

(Remarks to the Author)

Dielmann and coworkers report a photochemical approach for the autocatalytic degradation of SF₆ using mixtures of KOH, H₂O and i-PrOH. Aspects of the kinetics of the reaction, product distributions and mechanism as discussed in detail and supported by convincing experimental data and sensible discussion with reference to prior literature. My opinion is that this is a seminal contribution in sustainable fluorine chemistry and arguably one of the most important benchmark results in SF₆ degradation. I am fully supportive of publication. I have a few suggestions and requests for clarification that hope could improve the manuscript.

Photolysis: One potential drawback of the system is the need for high wavelength light source (280 nm). Can the authors provide more detail here in the manuscript. What is the combined power of these LEDs? Can we estimate quantum efficiency of the process and ensure that the system is not photon-limited at any point? Is there any tolerance for longer wavelengths and can they clarify if quartz equipment was used throughout or that reaction vessels are transparent across the necessary wavelength windows.

Mechanism: I wondered if the authors might be able to gain any insight into their mechanism by isotope effects. If D₂O is used instead of H₂O presumably we will see scrambling into the alpha position of iPrOH, if radical formation is reversible? There may also be a lack of KIE (independent experiments) if mass transfer is rate limiting as suggested. Is it also possible to independently generate the ketyl radical anion (e.g. under anhydrous conditions) and confirm it is a competent reductant for SF₆.

An alternative approach to scale up might be the use of flow apparatus where light penetration is more effective and salt products can be removed from the light source under flow. This is well beyond the current contribution but might be something for the authors to consider in the future.

Reviewer #2

(Remarks to the Author)

This article by Dielmann and coworkers describes UV-light promoted decomposition of sulfur(VI) hexafluoride in basic alcoholic solution. Recently, a number of studies focused on developing a method for degrading or valorizing this greenhouse gas. The authors provide some overview of these studies in the manuscript and SI, but, in my view, they missed several key publications, including electrochemical decomposition and valorization of SF₆. The system presented in this article is rather simple, and leads to relatively fast and inexpensive decomposition of sulfur(VI) hexafluoride that leads to the formation of inorganic salts (potassium fluoride and sulfite). Clearly, this is a useful process although there are doubts that adopting this to a ton-scale degradation of sulfur(VI) hexafluoride is an easier problem than relying on already existing thermal, photochemical or electrochemical methods. In addition, the reviewer doubts that the described chemistry is going to be of interest to the broad readership of Nature Communications. Although this article is well-prepared and written, it seems to be more suited for a specialized journal that is focused on environmental/green chemistry.

Reviewer #3

(Remarks to the Author)

In this manuscript, the authors present a photocatalytic strategy for the reduction of SF₆ using potassium hydroxide in water/isopropanol mixtures under UV irradiation. This approach demonstrates the potential for scalable and facile degradation of SF₆ under relatively mild reaction conditions. In addition, the authors propose a two-step process to explain the underlying autocatalytic mechanism. While the demonstration of SF₆ degradation under mild conditions is impressive, the discussion of the reaction mechanism remains vague and lacks sufficient clarity. Therefore, I recommend reconsideration of the manuscript after a major revision to address these mechanistic issues.

1. The authors appear to conflate the charge transfer process with electron detachment in their discussion (lines 150–167). Fundamentally, charge transfer refers to the redistribution of electrons within a molecular complex, typically from a donor to an acceptor, without changing the total number of electrons in the system. In contrast, electron detachment describes the process in which an electron is completely removed from a molecule or complex, resulting in a change in the total electron count. While there may be some correlation between the vertical detachment energy (VDE) and the excitation energy of a charge transfer process, the two are conceptually distinct. Therefore, the authors should clarify their mechanistic discussion using appropriate terminology, and to explicitly distinguish between electron excitation and detachment in both their experimental and theoretical analyses.

2. The authors should select appropriate theoretical tools to investigate the reaction mechanism. In principle, time-dependent density functional theory (TDDFT) is designed for calculating electronic excitation energies within the same number of electrons, and is not directly applicable to processes involving electron detachment. Therefore, using TDDFT to directly compute VDE, as done in the manuscript, is conceptually inappropriate. To properly model the system, the authors should include both the donor (OH⁻) and the acceptor (SF₆) in a combined molecular complex, such that the process corresponds to an intermolecular charge transfer. In this context, TDDFT would be an appropriate method to evaluate the excitation energy associated with electron redistribution. Additionally, it is strongly encouraged to explicitly consider solvent effects, as the coordination of solvent molecules to the reactants and products may significantly influence the charge-transfer energetics and overall mechanism.

3. Theoretical methods can play a more central role in elucidating the reaction mechanism, particularly by uncovering microscopic details that are inaccessible to experimental observation. Rather than merely reproducing the experimentally observed irradiation wavelength, computational studies should aim to provide mechanistic insights that guide and complement the experimental findings.

4. It is recommended that the authors unify the number of significant digits when discussing identical properties, even if the corresponding values are cited from different literature sources. For instance, the discussion of redox potentials in lines 179–182 lacks consistency in numerical precision. In addition, typographical errors should be corrected—for example, “Figure 2D” in line 209.

To conclude, the theoretical calculations, including methods, models and results, are unable to meet the requirements of the journal.

Version 1:

Reviewer comments:

Reviewer #1

(Remarks to the Author)

In my opinion the authors have now suitably address all the important points raised by referees. I am supportive of publication of the work in its current form.

Reviewer #3

(Remarks to the Author)

I would like to commend the authors for their revised manuscript, which successfully addresses most of the concerns raised in the first round of review. The revision has made significant progress in integrating the theoretical and experimental aspects of the study, particularly in terms of providing computational support to better understand the reaction mechanism. The authors' efforts to link the theoretical calculations to experimental results have strengthened the overall clarity and depth of the study, and their approach is now more consistent with a collaborative theory-experiment framework.

However, I still have a few points to highlight regarding the methodology and terminology. Specifically, when discussing charge transfer (CT), the calculations should include both the donor and the acceptor in the same model. While the authors have made improvements by including the OH⁻...SF₆ complex, the term "CTTS" in the SI is misleading, as it implies that solvent is part of the CT process. Based on the authors' description, the calculation corresponds to a vertical excitation of the donors (OH⁻, iPrOH, iPrO⁻), which serves as a primary criterion for determining whether the photocatalytic reaction, involving true charge transfer, can take place. However, this calculation does not represent the actual charge transfer process itself. To avoid confusion, I recommend replacing "CTTS" with a more accurate term such as "vertical excitation energy of the donor".

In summary, the manuscript now presents a much clearer picture of the reaction mechanism, and the theoretical calculations align well with the experimental data. Once the suggested terminology change is made, I believe the manuscript will be suitable for publication.

Reviewer #1

Dielmann and coworkers report a photochemical approach for the autocatalytic degradation of SF₆ using mixtures of KOH, H₂O and i-PrOH. Aspects of the kinetics of the reaction, product distributions and mechanism as discussed in detail and supported by convincing experimental data and sensible discussion with reference to prior literature. My opinion is that this is a seminal contribution in sustainable fluorine chemistry and arguably one of the most important benchmark results in SF₆ degradation. I am fully supportive of publication. I have a few suggestions and requests for clarification that hope could improve the manuscript.

Photolysis: One potential drawback of the system is the need for high wavelength light source (280 nm). Can the authors provide more detail here in the manuscript. What is the combined power of these LEDs?

Our response: Thank you for pointing this out. Details regarding the irradiation power of the LED array are included in the Supplementary Information (Chapter 2). Specifically, the LED-280-A device has a total irradiation power of approximately 220 mW. This information has now also been incorporated into the revised manuscript for clarity.

Can we estimate quantum efficiency of the process and ensure that the system is not photon-limited at any point?

Our response: Thank you for highlighting this important aspect. We conducted additional experiments to gain deeper insights into the photocatalytic system, which are summarized below. The results were added to the revised manuscript and SI.

Experiment 1: Systematic Variation of Irradiation Power (see the SI, Chapter 6.2.4)

To determine the transition from photon limitation to mass transfer limitation, we conducted experiments using the KOH/iPrOH system and irradiation setup 2 with varying numbers of LEDs (1, 3, 5, 10, and 20). The results (SI, Chapter 6.2.4) show that the SF₆ degradation rate increases with the number of LEDs up to 10 LEDs, but no further enhancement is observed with 20 LEDs. This indicates that the system is photon-limited when fewer than 10 LEDs are used but becomes mass transfer-limited at higher irradiation powers.

This interplay between photon and mass transfer limitations is specific to the irradiation setup and is critical for scaling up the system to a kilogram scale, which is currently under investigation in our laboratory.

Estimation of Quantum Efficiency Φ (see the SI, Chapter 8.9)

Since the SF₆ degradation process involves autocatalysis coupled with a redox catalysis chain reaction, we estimated the minimum quantum efficiency (Φ_{\min}). The detailed calculations are provided in SI (Chapter 8.9).

Without redox catalysis, a single photon can degrade a maximum of 2 SF₆ molecules. However, the experimentally determined SF₆ degradation rates (4.8 mmol/h for the KOH/iPrOH system and 7.5 mmol/h for the KOH/H₂O/iPrOH system) exceed the number of photons emitted by the LED (1.8 mmol/h). This confirms that the redox catalysis chain reaction significantly contributes to SF₆ degradation.

Experiment 2: Intermittent Irradiation to Assess the Contribution of the Dark Reaction (see the SI, Chapter 8.10)

To evaluate the role of the light-independent dark reaction (redox catalysis), we conducted two experiments (SI, Chapter 8.10):

(1): Continuous irradiation for 20 minutes resulted in an 85% yield.

(2): Intermittent irradiation (20 minutes total light exposure, with dark periods in between) resulted in a higher yield of 94%.

The higher yield in Experiment (2) indicates that redox catalysis contributes to SF₆ degradation during dark periods. However, it also shows that the redoxcatalysis lacks long-term stability and ceases shortly after irradiation stops. A possible explanation for this low turnover number is provided by DFT calculations on the

dissociation pathways of the SF₆ radical anion and the subsequent hydrogen transfer reaction (see answer to reviewer 3 below), which reveal that the generation of SF₅ radicals terminates the redoxcatalysis cycle.

Is there any tolerance for longer wavelengths and can they clarify if quartz equipment was used throughout or that reaction vessels are transparent across the necessary wavelength windows.

Our response: The wavelength screening described in the SI (Chapter 5.3) demonstrated that wavelengths longer than 280 nm do not result in SF₆ conversion. Since borosilicate glass efficiently absorbs light below 300 nm, we used quartz glass equipment for all irradiation experiments to ensure transparency across the necessary wavelength range. For comparison, we conducted an additional experiment using a borosilicate glass Schlenk tube instead of a quartz tube. The reaction rate was significantly slower (0.2 g/h compared to 0.7 g/h in the quartz tube), but complete SF₆ conversion was still achieved within 2 hours in both cases. These results have been included in the revised SI (Chapter 8.12).

Mechanism: I wondered if the authors might be able to gain any insight into their mechanism by isotope effects. If D₂O is used instead of H₂O presumably we will see scrambling into the alpha position of iPrOH, if radical formation is reversible? There may also be a lack of KIE (independent experiments) if mass transfer is rate limiting as suggested.

Our response: Thank you for suggesting this approach to gain further insight into the reaction mechanism. We conducted an irradiation experiment using the biphasic system with D₂O instead of H₂O. The reaction progress was monitored to investigate the presence of a kinetic isotope effect, and the organic phase was analyzed by ¹H, ²D, and ¹³C{¹H} NMR spectroscopy to detect deuteration of the reaction components. The detailed results are provided in the SI (Chapter 8.11).

Monitoring the SF₆ uptake revealed no KIE, as both systems exhibited identical reaction rates. This suggests that the reaction rate is not limited by hydrogen atom transfer but is instead limited by photon availability or mass transfer.

NMR analysis of the organic phase after the irradiation experiment showed the formation of HDO, isopropanol-d₁ (with deuterium incorporated into the hydroxy group, C_α-position, or C_β-position), and acetone-d₁. The deuteration of the hydroxy group of isopropanol and the formation of acetone-d₁ are expected under the basic conditions of the system. Notably, acetone is known to slowly incorporate deuterium at the C_β-position in the presence of D₂O, even under weakly basic conditions.

Consistent with the autocatalysis cycle, photoexcitation of acetone-d₁ generates two ketyl radicals, one of which carries deuterium at the C_β-position. This radical undergoes hydrogen atom transfer with a second isopropanol molecule, leading to the formation of isopropanol-d₁. Furthermore, the observed scrambling of deuterium into the α-position of isopropanol indicates that radical formation is reversible and confirms the in-situ generation of ketyl radicals.

Is it also possible to independently generate the ketyl radical anion (e.g. under anhydrous conditions) and confirm it is a competent reductant for SF₆.

Our response: To our knowledge, the dimethyl ketyl radical anion is not a stable or isolable compound. However, we conducted two experiments to investigate whether the in-situ generated species can activate SF₆. The detailed results are provided in the SI (Chapter 8.8).

In the first experiment, an NMR tube was charged with potassium metal (21.0 mg, 0.53 mmol) and THF (1 mL). Upon adding anhydrous acetone (0.04 mL, 0.53 mmol, dried over 3 Å molecular sieves), gas evolution was observed at the metal surface, and the THF solution turned slightly turbid, likely due to trace water in the acetone or slow decomposition of acetone. After 10 minutes, with potassium metal still present, the NMR tube was pressurized with 3 bar SF₆. This led to a color change from yellow to orange, more solid formation and the complete consumption of potassium. In the ¹⁹F NMR spectrum, the signal of SF₆ (δ = 57.3 ppm) was detected as the only fluorine-containing compound. The volatiles were evaporated, and the resulting solid was dissolved in water. Quantitative ¹⁹F NMR spectroscopy, using potassium triflate (10.0 mg, 0.05 mmol) as an internal standard (¹⁹F NMR resonances: δ = -78.6 (OTf⁻), -119.1 (F⁻) ppm) revealed the formation of fluoride: 3.5 mg (0.06 mmol).

In a control experiment, an NMR tube containing only potassium metal (21.0 mg, 0.53 mmol) and THF (1 mL) was pressurized with 3 bar SF₆. No visible reaction occurred after 1 day, and only SF₆ was detected in the ¹⁹F NMR spectrum ($\delta = 57.3$ ppm).

These results suggest that the in-situ generated species, likely the dimethyl ketyl radical anion, is capable of activating SF₆, whereas potassium metal alone does not react with SF₆ under the same conditions.

Reviewer #2

This article by Dielmann and coworkers describes UV-light promoted decomposition of sulfur(VI) hexafluoride in basic alcoholic solution. Recently, a number of studies focused on developing a method for degrading or valorizing this greenhouse gas. The authors provide some overview of these studies in the manuscript and SI, but, in my view, they missed several key publications, including electrochemical decomposition and valorization of SF₆. The system presented in this article is rather simple, and leads to relatively fast and inexpensive decomposition of sulfur(VI) hexafluoride that leads to the formation of inorganic salts (potassium fluoride and sulfite). Clearly, this is a useful process although there are doubts that adopting this to a ton-scale degradation of sulfur(VI) hexafluoride is an easier problem than relying on already existing thermal, photochemical or electrochemical methods. In addition, the reviewer doubts that the described chemistry is going to be of interest to the broad readership of Nature Communications. Although this article is well-prepared and written, it seems to be more suited for a specialized journal that is focused on environmental/green chemistry.

Our response: We agree that developing effective methods for degrading or valorizing SF₆ is of critical importance. Therefore, the field was recently summarized in several review articles (Refs. 6, 7, 9). However, to the best of our knowledge, and based on discussions with our industry collaborators, no established photochemical or electrochemical method currently exists for SF₆ disposal. If the reviewer is aware of such a method, we would be happy to reference it.

To ensure thoroughness, we re-examined the available references and included them in our comparative table of solution-based methods for SF₆ fragmentation in the SI (Chapter 10). Additionally, we have also compiled a second table summarizing the reported photochemical approaches (SI, Chapter 11).

Reviewer #3

In this manuscript, the authors present a photocatalytic strategy for the reduction of SF₆ using potassium hydroxide in water/isopropanol mixtures under UV irradiation. This approach demonstrates the potential for scalable and facile degradation of SF₆ under relatively mild reaction conditions. In addition, the authors propose a two-step process to explain the underlying autocatalytic mechanism. While the demonstration of SF₆ degradation under mild conditions is impressive, the discussion of the reaction mechanism remains vague and lacks sufficient clarity. Therefore, I recommend reconsideration of the manuscript after a major revision to address these mechanistic issues.

1. The authors appear to conflate the charge transfer process with electron detachment in their discussion (lines 150–167). Fundamentally, charge transfer refers to the redistribution of electrons within a molecular complex, typically from a donor to an acceptor, without changing the total number of electrons in the system. In contrast, electron detachment describes the process in which an electron is completely removed from a molecule or complex, resulting in a change in the total electron count. While there may be some correlation between the vertical detachment energy (VDE) and the excitation energy of a charge transfer process, the two are conceptually distinct. Therefore, the authors should clarify their mechanistic discussion using appropriate terminology, and to explicitly distinguish between electron excitation and detachment in both their experimental and theoretical analyses.

2. The authors should select appropriate theoretical tools to investigate the reaction mechanism. In principle, time-dependent density functional theory (TDDFT) is designed for calculating electronic excitation energies within the same number of electrons, and is not directly applicable to processes involving electron detachment.

Therefore, using TDDFT to directly compute VDE, as done in the manuscript, is conceptually inappropriate. To properly model the system, the authors should include both the donor (OH^-) and the acceptor (SF_6) in a combined molecular complex, such that the process corresponds to an intermolecular charge transfer. In this context, TDDFT would be an appropriate method to evaluate the excitation energy associated with electron redistribution. Additionally, it is strongly encouraged to explicitly consider solvent effects, as the coordination of solvent molecules to the reactants and products may significantly influence the charge-transfer energetics and overall mechanism.

Our response: The reviewer raises an important point. Initially, the calculation of VDEs were conducted to identify whether OH^- or isopropanol display excitations that align with the experimental setup (UV irradiation near 280 nm). The calculation points towards hydroxide showing indeed excitations in the correct range. In addition, the comparison of different methods and basis sets enabled us to identify the $\omega\text{B97XD/6-311++G(d,p)}$ level of theory as a suitable compromise between accuracy of results and computational effort for TDDFT calculations of larger systems in line with general recommendations in the literature.

The reviewer is of course correct that conclusion about a potential photo-induced charge transfer can only be obtained if both donor and acceptor molecules are considered in the calculation. Since the extension of the system from OH^- to the system of $\text{SF}_6\cdots\text{OH}^-$ leads to a dramatic increase in the electron count (from 10 to 80 along) and the associated number of atomic orbital basis functions (especially when using a split-valence triple-zeta basis set), the $\omega\text{B97XD/6-311++G(d,p)}$ level of theory was chosen based on the results obtained in the OH^- case.

A particular challenge was identifying suitable interaction motifs between OH^- and SF_6 due to the exceptionally weak interaction. Nevertheless, due to a series of extensive structure optimization both a hydrogen- and halogen-bonded interaction motif could be identified. Both represent genuine minima on the potential energy surface (i.e. only real vibrational frequencies).

TDDFT calculations of these combined $\text{SF}_6\cdots\text{OH}^-$ systems provided several key findings:

- In both cases the relevant excitations only showed a marginal shift compared in the relevant excitation from 278.5 nm obtained for isolated OH^- to 258.6 and 268.9 nm in case of the hydrogen- and halogen-bonded system, respectively. These numbers imply that the presence of SF_6 results only in a minor shift in wavelength that still aligns with the UV source employed in the experimental setup.
- Analysis of the involved molecular orbitals revealed that in both cases an excitation from the two highest occupied molecular orbitals (i.e. HOMO-1, HOMO) localized at the oxygen atom of OH^- to low-lying virtual states (i.e. LUMO, LUMO+1, etc.) associated to the SF_6 molecule can be identified. These results provide clear evidence in favor of a photo-induced intermolecular charge transfer process as initially proposed.
- The excitation is to a large extent independent of the orientation of OH^- relative to SF_6 as indicated by the similar wave lengths of 258.6 and 268.9 nm associated to comparably large oscillator strengths of 0.0854 and 0.0732 in case of the hydrogen- and halogen-bonded system, respectively.

This information has been compiled in a new figure in the Supporting Information (Figure S68) and is discussed in Chapter 12.2.1.

All these results have been obtained in implicit solvation using the PCM (polarizable continuum model) approach in conjunction with the appropriate settings for '2-propanol'.

While we share the opinion of the reviewer that an explicit treatment of solvation effects is highly desirable and we have actively pursued this avenue, the required number of solvent molecules and the associated increase in computation time (even when reducing the basis set to double-zeta level) proved too demanding for the computational resources available to the institution.

In order to suitably expose the combined $\text{SF}_6\cdots\text{OH}^-$ system to a single layer of solvent a minimum of twelve isopropanol molecules are required, increasing the total number of atoms to 153 (488 electrons, 1650 basis functions with about 2600 primitive Gaussians at double-zeta level only). Unfortunately, such as systems is

beyond the capacities available for the theoretical group. If the structure were rigid (e.g., a tightly coordinated inorganic complex with few accessible degrees of freedom), such a calculation might be just barely feasible. However, the exceptionally weak interaction between SF₆ and the aliphatic residues of isopropanol makes the individual molecular units highly mobile. This in turn leads to highly ineffective structure optimizations in addition to the extraordinary long time required for an SCF cycle. Even a reduction of the solvent molecules close to SF₆ to only three suffers from similar problems and ultimately, the calculation had to be aborted.

We also tested simplified QM approaches for pre-equilibration such as semi-empirical density functional tight binding (DFTB). Despite the availability of 3rd order DFTB parameters for the interactions in the set {C-H-O-S-F} the parameterization does not provide a suitable treatment of the SF₆...OH⁻ system. As a consequence, the configuration of SF₆ and OH⁻ had to be kept constrained to the DFT minimum in the energy minimization (*i.e.* only the solvent was subject to optimization).

The image below shows the last configuration obtained using this DFTB-based pre-equilibration for the system considering twelve added isopropanol molecules:

Subjecting this particular structure to a TDDFT test calculation immediately failed due to the associated demand in memory. Thus, for multiple reasons, the requested calculations considering explicit solvation can unfortunately not be carried out with the available computational equipment. In addition to the limitation in memory, this does not consider the potentially very high runtime associated *a)* to the system size and *b)* the potentially large number of excitations associated to the OH groups of the twelve isopropanol molecules, which in turn will require the calculation of much larger number of excited states.

While partial solvation (e.g. just adding isopropanol to OH⁻) could be an option to reduce the computational demand, it will introduce asymmetries in the solvation. Likewise, a mixed approach combining an implicit and explicit treatment of solvent effects proved highly unreliable in calculations carried out at our department in the past.

Nevertheless, even if this entire workflow could be realised within a time window of several months, it would only represent a single configuration out of a high number of potential microstates of the solvated ensemble, providing only a single snapshot of the system that could very well lead to misleading results. The most suitable approach to study the ensemble would be a clustering approach based on a QM/MM-type molecular dynamics simulation trajectory. However, to the best of knowledge, an adaptive QM/MM MD simulation of a highly complex solute in isopropanol has never been attempted. From our own experience in adaptive QM/MM MD, which

includes the simulation of the much simpler CO₂ molecule in dichloromethane (<https://doi.org/10.1016/j.molliq.2022.119840>), such a simulation is at present highly unfeasible.

We hope the reviewer agrees to our long discussion, highlighting that while a consideration of explicit solvation of the SF₆⋯OH⁻ system is at present limited by computational resources, the treatment of the combined SF₆⋯OH⁻ system in implicit solvent including orbital analysis provided manifold insight into the properties of this highly complex interaction that are in line with the observation by the experimental colleagues.

3. Theoretical methods can play a more central role in elucidating the reaction mechanism, particularly by uncovering microscopic details that are inaccessible to experimental observation. Rather than merely reproducing the experimentally observed irradiation wavelength, computational studies should aim to provide mechanistic insights that guide and complement the experimental findings.

Our response: Our response: The reviewer raises important point. In addition to the extensive calculations discussed above under point 2) several additional DFT calculations (ωB897XD/6-311++G(d,p)/PCM) associated to the thermodynamic properties of key reaction steps in the catalytic cycles have been carried out. These involve (all energies in kJ mol⁻¹):

- The decomposition of SF₆ radical anion:

	ΔE	ΔG
SF ₆ ^{•-} → SF ₅ ⁻ + F [•]	134.0	103.1
SF ₆ ^{•-} → SF ₅ [•] + F ⁻	140.4	118.2

- Proton abstraction at isopropanol:

	ΔE	ΔG
F [•] + (CH ₃) ₂ CHOH → HF + (CH ₃) ₂ COH [•]	-164.1	-189.1
SF ₅ [•] + (CH ₃) ₂ CHOH → HSF ₅ + (CH ₃) ₂ COH [•]	51.6	51.0

In all cases minima were verified via the calculation of the associated vibrational frequencies, that are also required for the thermochemical analysis to obtain ΔG-values.

These data of individual reaction equilibria yield valuable information to interpret the experimental findings and have been discussed in the manuscript and the associated supplementary material as appropriate (Chapter 12.2.2 and 12.2.3).

4. It is recommended that the authors unify the number of significant digits when discussing identical properties, even if the corresponding values are cited from different literature sources. For instance, the discussion of redox potentials in lines 179–182 lacks consistency in numerical precision. In addition, typographical errors should be corrected—for example, “Figure 2D” in line 209.

Our response: This has been corrected.

Reviewer #3

However, I still have a few points to highlight regarding the methodology and terminology. Specifically, when discussing charge transfer (CT), the calculations should include both the donor and the acceptor in the same model. While the authors have made improvements by including the $\text{OH}^- \cdots \text{SF}_6$ complex, the term "CTTS" in the SI is misleading, as it implies that solvent is part of the CT process. Based on the authors' description, the calculation corresponds to a vertical excitation of the donors (OH^- , iPrOH , iPrO^-), which serves as a primary criterion for determining whether the photocatalytic reaction, involving true charge transfer, can take place. However, this calculation does not represent the actual charge transfer process itself. To avoid confusion, I recommend replacing "CTTS" with a more accurate term such as "vertical excitation energy of the donor".

Our response: We appreciate the reviewer's valuable comment and have revised the terminology accordingly, as suggested.